# Holocene man-occupied caves and transformed wetlands as facilitating factors for *Leishmania infantum* in South America

Sérvio Pontes Ribeiro[1,2/+], Camila de Paula Dias[1,2], Rafael Vieira Duarte[2,4],
Maria Fernanda Brito de Almeida[2], Luccas Gabriel Ferreira Malta[1], José Dilermando Andrade Filho[3],
Alexandre Barbosa Reis[4], Marcos Horácio Pereira[1], Carlos EV Grelle[5], Jesus G Valenzuela[6],
Tiago Donatelli Serafim[6], Nelder Figueredo Gontijo[1]

[1]Universidade Federal de Minas Gerais, Instituto de Ciências Biológicas, Departamento de Parasitologia,
 Laboratório de Fisiologia de Insetos Hematófagos, Belo Horizonte, MG, Brasil
[2]Universidade Federal de Ouro Preto, Instituto de Ciências Exatas e Biológicas, Núcleo de Pesquisa em Ciências Biológicas,
 Laboratório de Ecologia do Adoecimento e Florestas, Ouro Preto, MG, Brasil
[3]Fundação Oswaldo Cruz-Fiocruz, Instituto René Rachou, Grupo de Estudos em Leishmanioses, Belo Horizonte, MG, Brasil
[4]Universidade Federal de Ouro Preto, Escola de Farmácia, Departamento de Análises Clínicas, Núcleo de Pesquisa em Ciências Biológicas,
 Laboratório de Imunopatologia, Ouro Preto, MG, Brasil
[5]Universidade Federal do Rio de Janeiro, Departamento de Ecologia, Rio de Janeiro, RJ, Brasil
[6]National Institutes of Health, National Institute of Allergy and Infectious Diseases, Vector Molecular Biology Section,
 Laboratory of Malaria and Vector Research, Rockville, MD, USA

**BACKGROUND** In the Holocene, South American humans transformed large extensions of the continent, especially in cave ecosystems. Such transformations produced predictable eutrophic habitats that could have attracted foxes and further favored the adaptation of *Lutzomyia longipalpis*, insect vector of American Visceral Leishmaniosis (AVL), to human-contaminated habitats.

**OBJECTIVES** Here we present spatial analyses on the distribution of caves, Holocene human populations, the present-days main wild reservoirs of *Leishmania infantum*, *Cerdocyon thous* and *Lycalopex vetulus*, and the vector *Lu. longipalpis* in Brazil.

**METHODS** The presence or absence of *Lu. longipalpis* in function of cave abundance, based on coordinates of all recorded samples (Fiocruz and GBIF database and literature), and cave locations taken from ICMBio/CECAV database, were tested by contingency table. The overlap in the distribution of *Lu. longipalpis*, *C. thous* and *L. vetulus* with humans from Holocene was tested by permutational multivariate analysis of variance (PERMANOVA) from a nonmetric multidimensional scaling (NMDS), using published archaeological data on human demography and ICMBio's data on caves and foxes.

**FINDINGS** Caves are present in 18% of Brazilian municipalities, and *Lu. longipalpis* were significantly more frequent in these places than in municipalities without caves. Native humans and foxes have broader distributions than caves but co-occurred with *Lu. longipalpis* in cave-ecosystems.

**MAIN CONCLUSION** The most relevant implication of our findings is that visceral leishmaniasis should be considered a long-term human related disease, associated with few sandfly species well adapted to our modified, and heavily contaminated, environments.

Key words: parasite-responsive landscapes - American Visceral Leishmaniasis - disease ecology - insect vectors - *Lutzomyia longipalpis* - canids

Human infectious diseases and parasitism have travelled with people, but with different chances to invade new human populations and habitats. The key to a successful environmental spillover depends on the probability of transmission and host susceptibility,[1] but also on how favorable the new environment might be to accommodate ecological demands of a likely invasive parasite. The European human invasion of the Americas introduced a substantial number of new parasites and pathogens to the continent, and some became part of the global pattern of disease distribution.[2] However, the ecological conditions that were most favorable for successful parasitism colonization have never been properly explored.

Host species spatial distribution, abundance and habitat fidelity are key factors in host-parasite ecological dynamics, affecting transmissibility and evolution of virulence.[3] In scenarios where tolerance (being infected without getting sick) is more likely to evolve than resistance (not getting or maintaining the infection), would have a positive effect on a parasite population that has spilled over, allowing it to spread silently. In cases of a recurrent and predictable interaction between host and par-

Financial support: This work was supported by the CNPq (Grant/Award Numbers: 303680/2020-2, 442499/2023-0 and 402949/2022-6), FAPEMIG (Grant/Award Number: PPM-00792-18. Project "Applications of genomics in the context of OneHealth" - identifier RED-00181-23). This research was supported in part by the Intramural Research Program of the National Institutes of Health (NIH).
+ Corresponding author: serv;opr@gmail.com | ⓘ https://orcid.org/0000-0002-0191-8759

**Handling editor:** Elisa Cupolillo | ⓘ https://orcid.org/0000-0002-0620-3250

asite, tolerance of a proportion of the host genotypes may boost parasite specialization.[4,5] Such a pattern is typically associated with plant-pathogen systems, but animals with low vagility and high habitat fidelity could produce a similar evolutionary scenario.[5] As settled and landscape-attached human communities, we became a long-term predictable host, and this may have been a driver of parasite spillovers, namely of those parasites introduced by migrant humans into native humans and animals.

The probability of encountering a susceptible host might favor generalism in an ecological community dominated by either spatially unpredictable, highly vagile or rare host species.[5,6] Conversely, the predictability of successfully finding a susceptible host could result in specialization to one or a few hosts species.[7] Applying this concept to vector-borne diseases, the prevalence among humans depends, first, on the distribution of the vector species. Hematophagous insect dispersal is driven by habitat suitability and host density and distribution.[8] Possible trade-offs between surviving in certain environments (especially for immatures) and optimizing egg production (depending on finding predictable blood sources) might affect oviposition choices, dissemination behaviors, as well as the evolution of host species range.[8,9] This combination of optimal habitats and the predictability of human settlements may have been shaping the evolution of human parasites since we abandoned nomadism.

A vertebrate animal species regularly associated with a place, such as dens and territorial dwellings, leaves environmental signals to hematophagous insects about their location. In the case of humans, predictability could be a main driver for insect vector of diseases to specialize in anthropomorphic environments and domestic animals.[7,10] Further, a disease transmitted by insect vectors with a highly specialized infection route, and with a substantial proportion of asymptomatic hosts, is likely a well-established specialization.[3,11] The above description fits the leishmaniasis, especially the visceral form found in wild canids, domestic dogs and humans, in both the Old and New World.[12,13]

An aspect rarely discussed in visceral leishmaniasis ecology is that human and other large animals' interactions might have created richer and more predictable microhabitats for sandfly immatures. However, a limited number of sandfly species can transmit visceral leishmaniasis in both Old and New World, compared to cutaneous leishmaniasis.[13] These are the same few sandfly species associated with modern densely human populated suburbs. Ecological filters such as highly toxic or polluted environments found around our settlements could reduce the chance of many species associating with us.[14]

Many distinct canid species have experienced convergent domestication processes throughout the history of humanity, dating back 40,000 years ago (y.a. hereafter) in North America,[15,16] but also in Egypt (7,000 y.a.), Iraq (8,000 y.a.), Cyprus (5,500 y.a.), Greece (6,000 y.a.)[17,18] and also in South America (2,000-3,000 y.a.)[19,20] In South America, the 10 living canid species are spread out along the whole continent, and are descendants of a sole ancestral species, which invaded the continent between 3.9 million years ago (Mya hereafter) and 3.5 Mya,[21]

just before the sandfly genus *Lutzomyia* is supposed to have arrived, both through the Panama Isthmus.[13,21] In South America, American visceral leishmaniasis (AVL hereafter) is caused mostly by *Leishmania infantum*, and transmitted by *Lutzomyia longipalpis* (Psychodidae; Phlebotominae). Two fox species in South America, *Cerdocyon thous* and *Lycalopex vetulus* are listed as reservoirs of *L. infantum*.[22,23] Regardless ongoing debate, the most accepted hypothesis is that at least part of AVL is caused by a sub-population of *L. infantum* emerged from a Portuguese population, likely arrived around the colonization time, 500 y.a.[24]

Considering a spillover of *L. infantum* from Old World dogs to South American fox species implies that the common vector, *Lu. longipalpis*, could have blood-fed on recent invaded domestic dogs, as well as associated to their domestic environment. A recent study has shown that domestic dog feces are a suitable substrate for the full development of *Lu. longipalpis*.[25] Canid dens are composed of feces and prey corpses under distinct decaying stages, resulting is a very special organic matter-rich microhabitat.[26]

On the other hand, Forattini[27] described *Lu. longipalpis* as a species constrained to rocky-shelters, caves or deep creek valleys within the savanna or dry forest Biomes. Hence, this species is constrained to very wet and predictable microhabitats in ecosystems submitted to a seasonal climate, with very dry winters, when low relative air humidity predominates. Whether a cave also provided shelter for foxes and humans in the past, it became a wet but also nutrient-rich environment. Therefore, a resource and climatic favorable and predictable microhabitat.

South America had a fast and diverse dispersion of humans from 13,000 y.a.,[28,29] followed by a stable, relatively low population around a carrying capacity from 9,000 to 5,500 y.a.[30] Most of these stable occupations were along the coast, in the Amazon or in the cave-rich landscapes of central highlands.[30,31] Later, from 5,5000 to 2,0000 y.a., new waves of migrations, combining a complex movement of linguistic groups spread and mixed with original groups. First, Jê-speaking groups, which mixed and changed Sambaqui-builders' traditions in the coast.[31] After, the Tupi-Guarani populations, which occupied and transformed most of the large riverbeds and wetlands of the continent.[30,31,32] An unknown aspect is whether such largely human-transformed landscapes could favor population growth of both vectors and reservoirs of parasitic human diseases.

The fast change in the carrying capacity from 5,500 y.a., followed by an exponential growth of continental and coastal populations, could only be possible by technological changes in the management of the landscape, most likely related to a new wave of Tupi-Guarani and related groups around the continent.[30,33] Such land management with fast population growth was most likely related to sedentarism, and overlapping with previously existent cave people.[28] Such possible scenario must have caused sensitive transformations in the habitats, making them more eutrophic, as usually happen to human settlements. Therefore, the environment also may have become attractive to commensal canids,[16] includ-

ing South American foxes.[20,34] Whether habitat enrichment and the constant presence of these interactive hosts might have driven *Lu. longipalpis* to adapt for living near humans is currently under investigation.

Neither the existence of pre-European complex societies, densely distributed human populations,[28,30] nor the indigenous likely domestication (or at least taming) of foxes[20] were taken into consideration to describe the natural history of visceral leishmaniasis in South America. Assuming the most likely scenario, that *L. infantum* invaded the continent with Portuguese infected dogs, there is a lack of studies on how such spillover to native foxes and people may have happened. To understand whether there was pre-colonial human-made niche for this parasite to invade could help to understand how this disease later migrated to suburbs of modern cities in the continent.

Here we explored the hypothesis that *Lu. Longipalpis'* distribution may overlap with the distribution of caves occupied by human populations from 9,000 y.a., being its present-days distribution a trace left from a pre-historic ecological association. Further, we also evaluated whether fox species which are implicated as wild reservoirs of AVL, overlap with *Lu. longipalpis* and early South America human distribution. We further discuss the present distribution of the disease and *Lu. longipalpis* in wild areas that were, until recently, the home of large native human populations and explored ecological paths for this sandfly to have invaded urban ecosystems.

## MATERIALS AND METHODS

*Study area* - South America has a diverse and abundant number of caves.[35] According to governmental survey, Brazil has documented caves in 995 municipalities (Available from: https://www.gov.br/icmbio/pt-br/assuntos/centros-de-pesquisa/cecav/), which correspond to 18% nationwide number of municipalities. Rocha et al.[36] pointed out a total of 16,089 known caves, 68% of them are limestone caves (Brazilian Karst), and 49.7% found in the savanna (Brazilian Cerrado) vegetation, where *Lu. longipalpis* predominates, along with the two fox species (*C. thous* and *L. vetulus*) incriminated as reservoirs for AVL. Likewise, according to Goldberg et al.[30] kernel density map of humans during the Holocene in South America (13,000 to 4,000 BP), their occurrence overlapped quite substantially with biogeographic regions with an abundance of caves.

Although in modern times most *Lu. longipalpis* samples are found in urban areas,[37] we hypothesize them to also happen in urban locations with similar ecological conditions of cave hidden den-like habitats, *i.e.*, city locations with low to no sanitary infrastructure with spots of soil and water contaminated with rich organic matter, from dog feces to open air sewage. Although any urban's neglected, wet, dirty, and sheltered dwelling spaces can provide ideal breeding grounds for this species, low-income neighborhoods in the vicinities of cave-rich ecosystems might have greater chances of having long-term endemicity of AVL.

To test this hypothesis, we first verified whether *Lu. longipalpis* distribution might be restricted to those biogeographic regions rich in caves, testing presence or absence of the sandfly in function of cave abundance. We took the coordinates of all recorded samples of *Lu. longipalpis* in the Fiocruz database (Available from: https://specieslink.net/search/ samples from 1956 to 2015), double checked in the free and open access to Biodiversity database (Available from: https://www.gbif.org/). These two databases comprise the majority of scientific sampling of *Lu. longipalpis* in the country. Other samples from health service were not used as identification precision may vary excessively. On the other hand, cities with confirmed occurrence of the species in the literature and not recorded by the Fiocruz's or GBIF database were also added (taking the core coordinate of the city) to decrease database record bias. The known cave locations and estimated locations were taken from ICMBio/CECAV database (Available from: https://www.gov.br/icmbio/pt-br/assuntos/centros-de-pesquisa/cecav/).

We assumed municipality as territory unit, but also, we corrected by neighbor caves. For each occurrence of *Lu. longipalpis*, we associated the number of caves in that municipality, and then checked by actual cave proximity, matching each sampling location with any actual cave occurrence within a radius of 100 Km from the center in the coordinates of *Lu. longipalpis* sample point, using Google Earth distance tools. This area size was defined arbitrarily, but in order to deal with the wide range of municipality sizes in the Country [from 3.5 to 60,431 Km$^2$ in Southeast and Northeast regions (Available from: https://www.ibge.gov.br/geociencias/)]. However, this correction was made only to prevent any underestimation of cave density in situations in which a *Lu. longipalpis* sample could have been taken in a municipality out of core distribution of caves, but still close enough to cavities. Then, caves found in neighboring municipalities within this radius were summed up for that sandfly occurrence.

In order to avoid inflating the zeros, we excluded the municipalities belonging to Biomes unlikely to have *Lu. longipalpis*, namely the wet Amazon forests and subtropical South of Brazil. These are also regions with irrelevant public records of AVL. All remaining Brazilian municipalities with no sampling or confirmed records of *Lu. longipalpis* were added as a "zero" occurrence. Because a reasonable proportion of zeros might be due to lack of research rather than actual absence of the species, we validated the distribution of *Lu. longipalpis*, and verify the effect of possible under sampling, by overlapping the distribution of the insect with the distribution of AVL incidence of cases in the Country, taken from the Information System of Injuries Notification (SINAN), Ministry of Health, from 2001 to 2021.

We then tested the overlap in the distribution of *Lu. longipalpis*, *C. thous* and *L. vetulus* with humans from Holocene. Holocene human distribution came from data made available in Goldberg et al.[30] The foxes species data were obtained from ICMBio/Sistema de Avaliação do Risco de Extinção da Biodiversidade - SALVE (Available from: https://salve.icmbio.gov.br/) - accessed in 28/04/2023.

Kernel density maps were produced for each species based on the distribution in the database mentioned above, and on the Goldberg et al.[30] published database.

All distributions were overlap with projections of high potential for the occurrence of caves in Brazil, found in Jansen et al.[35] Maps were produced in Arcmap 10.3 using default search radius for sandfly and foxes, and a search radius of 660 Km for Holocene humans.

*Lutzomyia longipalpis* distribution in function of proximity of caves was tested with a Contingency Table using MINITAB v21.3. The co-occurrence of the four species (humans, the two foxes and *Lu. longipalpis*) and the caves were tested by permutational multivariate analysis of variance (PERMANOVA). For such, we built a nonmetric multidimensional scaling (NMDS) analysis for all four species together and also for *Lu. longipalpis* and humans separate, using PAST v4.13. Using a simple dissimilarity matrix-based and non-metric method, it is possible to take coordinates as environmental variables and so scale up species by similarity in their location.[38,39] The PERMANOVA compares groups of species and tests whether their distribution centroids and data scattering in the multidimensional space were significantly different.[38,39]

## RESULTS

*Lutzomyia longipalpis* was found in only 65 municipalities. Regardless apparent under sampling, these sites were widespread all over the country. The species was significantly more recorded in municipalities with caves or at least 100 Km near a cave in a neighboring municipality (Chi-Square = 21.58, p < 0.0001, d.f. = 1). Within the biogeographic range of present days AVL incidence, which is mostly in the Biomes of Cerrado, Atlantic Rainforest, and Dry Forests, municipalities with mapped caves were 767, summing up 18% of total. On the other hand, 86% of database-deposited records of *Lu. longipalpis* came from these same 767 municipalities with caves.

Conversely, from all municipalities with caves (882), 70.2% of them had recent records of AVL cases, which composed 21.3% of the total municipalities with AVL (2910 out of 5565 Brazilian municipalities). Still, the majority of the 52% Brazilian municipalities which had AVL in present days happened within the Biomes above cited.

Also, this sandfly species showed a significant overlap distribution with Holocene humans and the two foxes species incriminated as reservoirs of *L. infantum*, *C. thous* and *L. vetulus* (Fig. 1). Both foxes′ distribution were far broader than *Lu. longipalpis* or Holocene humans, as they occupy a much larger territory, besides the cave-rich biogeographic regions (PERMANOVA F = 3.34, p <0.007, Fig. 2).

Then, the exclusion of the foxes from the NMDS/ PERMANOVA analyses exposed a non-significant difference in the distributions of present days *Lu. longipalpis* and Holocene humans and caves (PERMANOVA 0.51, p > 0.5, Fig. 2). Still, the distribution of all four species strongly suggest that coexistence must have happened among them in many locations in the central Brazil, with two especially strong hotspots, in interior Northeast Region and Central Minas Gerais State, and for quite a long time (Fig. 3).

## DISCUSSION

Visceral leishmaniosis is a disease that evolved through an ecological path completely different from the cutaneous form. While cutaneous leishmaniasis is caused

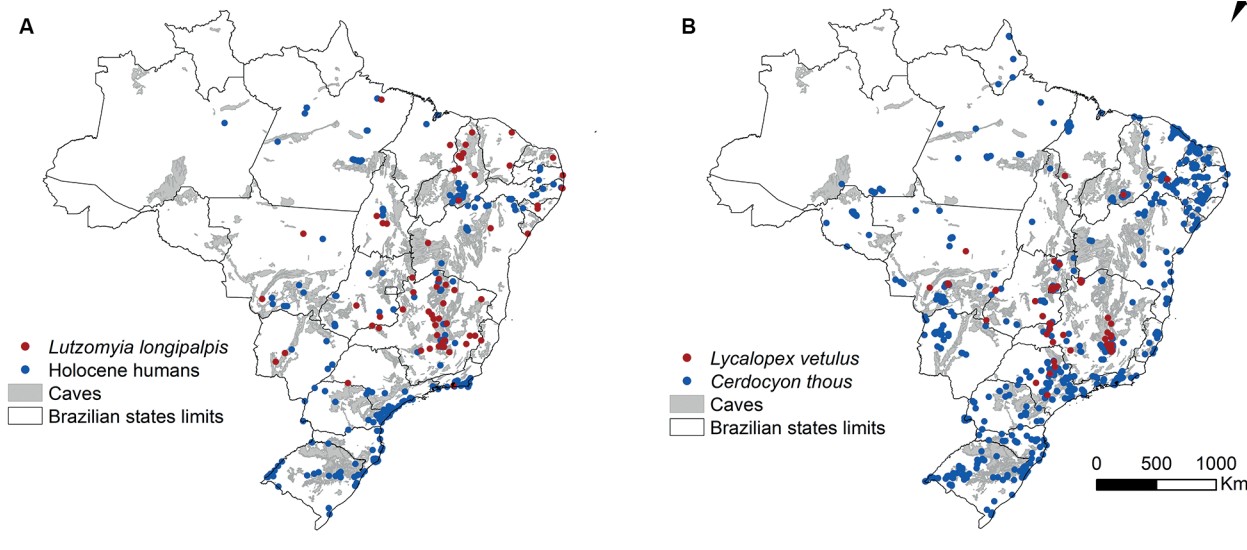

Datum: SIRGAS 2000
Projection: Geografic Coordinate System (South America)

Fig. 1: distribution of Holocene humans, *Lutzomyia longipalpis* (A), *Cerdocyon thous* and *Lycalopex. vetulus* (B) overlap with the expected distribution of caves in Brazil. Coordinates of all recorded samples of *Lu. longipalpis* in the Fiocruz database (Available from: https://specieslink.net/search/ samples from 1956 to 2015), double checked in the free and open access to Biodiversity database (Available from: https://www.gbif.org/), and the known cave location and estimated locations from Instituto Chico Mendes de Conservação da Biodiversidade/Centro Nacional de Pesquisa e Conservação de Cavernas (ICMBio/CECAV) database (Available from: https://www.gov.br/icmbio/pt-br/assuntos/centros-de-pesquisa/cecav/). The foxes data were obtained from ICMBio/Sistema de Avaliação do Risco de Extinção da Biodiversidade - SALVE (Available from: https://salve.icmbio.gov.br/), accessed in 28/04/2023. Holocene human distribution came from available data in Goldberg et al.[30]

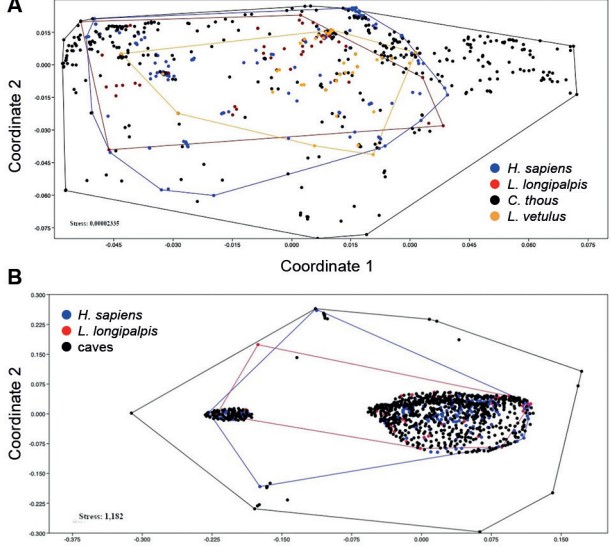

Fig. 2: non-metric multidimensional scaling (NMDS) from coordinates of occurrence of Holocene humans, *Lutzomyia longipalpis*, *Cerdocyon thous* and *Lycalopex vetulus* (A) and occurrence of Holocene humans, *Lu. longipalpis* and caves (B). The coordinate axes are collapsed information from original multivariate factors, in this case, the geographical actual coordinates for each location where species were recorded. These combined coordinates created by rank orders allow statistical comparison of co-occurrence.

by more than 20 *Leishmania* species (Available from: https://www.cdc.gov/dpdx/leishmaniasis/index.html)[13] VL is caused by only a few species. The most common are *L. infantum*, *L. donovani*[13,40] and, sporadically, *L. tropica*[41] and *L. amazonensis*.[42] There are also quite a few sandfly species capable of transmitting *L. infantum* or *L. donovani*. In the New World: *Lu. longipalpis*, secondarily *L. cruzi*,[43] and sporadically found in other species such as *Migonemyia migonei*, *L. evansi*, *Nyssomyia whitmani* or species of *Evandromyia*.[13,44,45,46,47] In the Old World: *Phlebotomus perniciosus*, *P. major*, *P. chinensis*, *P. longicuspis* and few other sporadic species.[48,49]

The geographic distribution of *Lu. longipalpis* and AVL reservoir foxes in Brazil showed a clear association of these species with cave habitats where, in the past, human had a consistent occupation. For instance, the oldest South America skeleton, Luzia, 10,000-year-old women, was found in a cave in Lagoa Santa Municipality, Minas Gerais State, a region abundant in caves and endemic for AVL. Fox dens are found in the crevices and holes around and in the entrances of caves. Nearby Luzia's cave (Lapinha cave), we sampled *Lu. longipalpis* females around feces in a fox den (Fig. 4). For the crab-eating fox, *C. thous*, the proximity to humans seemed not to be a problem even in present days, but an opportunity of resources, as described for Indian free ranging dogs.[50]

*Lutzomyia longipalpis* invaded South America by the same time the ancestor canid, likely a direct ancestor to the *Lycalopex* genus or a *Cerdocyon* species.[13,21] The accepted scenario is that canids spread thru the continent following xeric habitats, as savannas (and, thus, an ecological filter), by the time the Panamanian connection was formed between North and South America.[21] A savanna route could have been decisive for the widespread and radiation of most South American canids around 1.0 Mya, when dry Pleistocene periods made savannas predominant in the whole continent.[21,51] The existence of some level of mesic microhabitats around dens, eventually within gallery/deep valley forests, would have been a very favorable habitat for *Lu. longipalpis* immature development.[27,52]

Humans arrived much later, and likely due to climate unpredictability during the Younger Dryas (from 13,074 to 11,775 y.a.), started to occupy sheltered habitats as caves.[28,53] Peopling around caves were intensified in early Holocene, from 10,000 to 9,000 y.a.[28] Thus, our hypothesis is that humans stepped into this fox-sandfly habitat, transformed it into a much more eutrophic, plenty of predictable organic matter spots, to which *Lu. longipalpis* may have quickly adapted.

As humans and foxes are carnivores, an environment abundant in iron-rich feces and prey corpses under distinct decaying stages are a likely output for this interaction, as already recorded by other hunter gathering populations and canid dens.[16,20,26] Thus, they might produce a blood-rich organic matter, highly contaminated by iron, especially due to Heme group decomposition, which releases $Fe^{+3}$, a more toxic form of this element. Hence, human and fox interaction could produce an environmental filter for decomposers to overcome, a likely explanation for the existence of so few sandfly species associated with them. Ribeiro[54] showed that *Lu. longipalpis* has oviposition preference for decaying rat corpses and fresh dog feces, if compared with usual substrates tested in the literature. Also, it was found that this species can complete its life cycle on such substrates, showing an eco-evolutionary basis for this sandfly proximity with city suburbs.

Tupi-Guarani populations expanding towards cave-rich biogeographic regions, coming from mid-Amazonia, and then overlapped in time and space with original cave populations,[32] may have created ecological conditions for *Lu. longipalpis* to follow humans up to a new, technologically transformed habitat. Intense human constructions for ritualistic burial, such as the coastal Sambaquis, must have created an extra habitat very rich in decaying corpses and wet microhabitats. Most of these burials were sequential, resulting in a great number of bodies close together in decomposition, summed up to offerings of dead animals,[55,56] which must have resulted in organic matter-rich habitats.

A location with a well understood native civilization history, and where AVL cycle was properly studied is the Marajo Island.[12] The Marajo is an immense 40,100 Km$^2$ Island formed by the delta of the Amazonia River. It is a floodplain covered by savanna vegetation, and it was originally occupied by a complex native society, quickly exposed to European invaders in the early XVI Century, due to the proximity to Belém city.

Lainson & Rangel,[12] after Lainson et al.,[23] supported the hypothesis that *C. thous* was the sylvatic reservoir of the AVL, which infected dogs, and then humans, in small rural farms close to primary forests. This was a well-accepted hypothesis, built from a sample in the

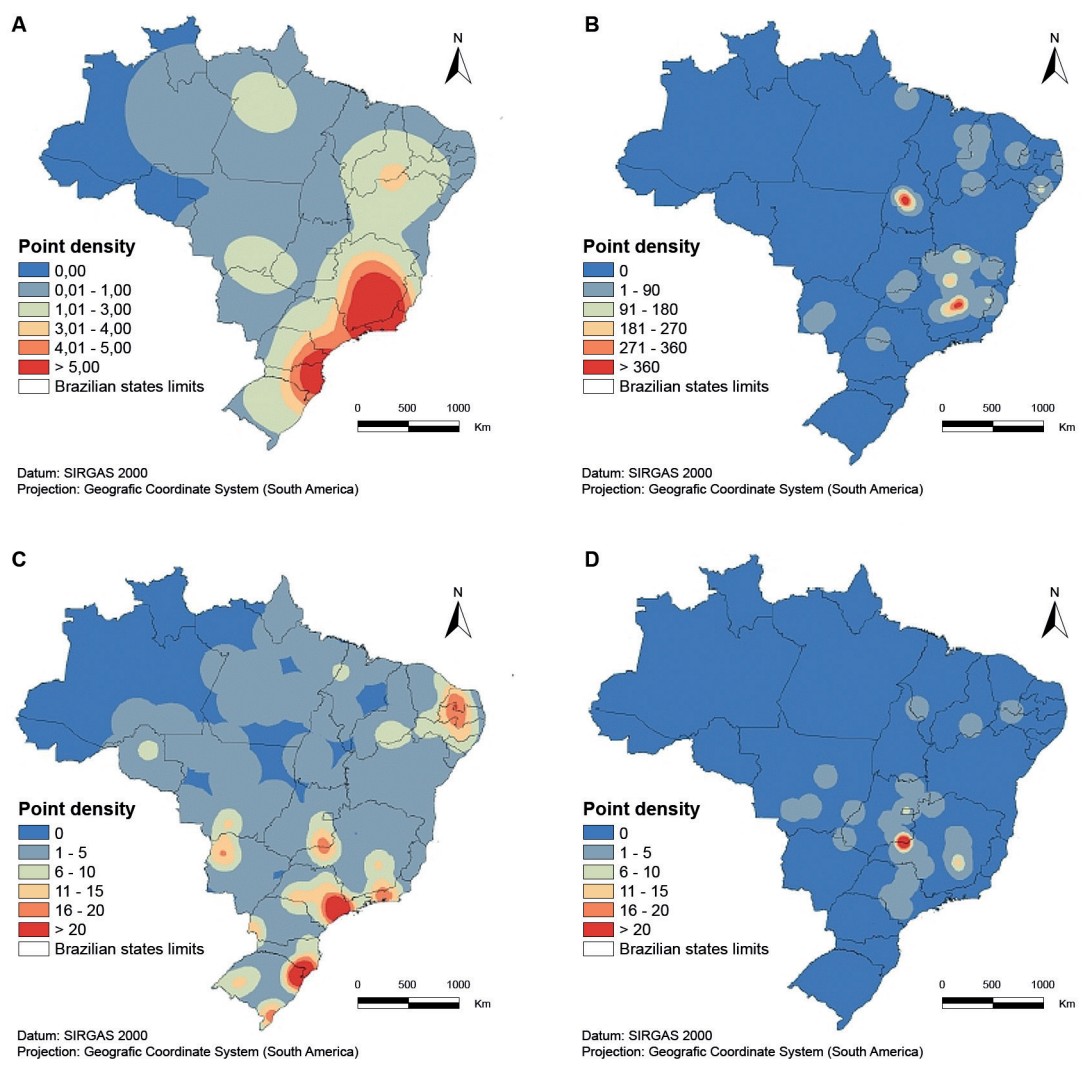

Fig. 3: kennel maps for the Holocene humans (A), *Lutzomyia longipalpis* (B), *Cerdocyon thous* (C) and *Lycalopex vetulus* (D) overlap with the expected distribution of caves in Brazil. Point densities show the regions with the highest to the lowest occurrence of each studied event.

Marajo Island, in Salvaterra village, a location near a so-called primary forest. However, being the Island highly populated by indigenous before Europeans colonization, the possibility of a pristine, primary forest is remote.[57] Here we offer an alternative interpretation to Lainson's pioneering and important findings.

In 5,500 y.a., the ancestral of Jê and Tupi-guarani groups, especially the latter, were able to deal with agriculture in the semi-arid, seasonal environments, and occupied wetlands from the Pantanal Matogrossense to Marajo Island.[30,32,58] These people built mounds of large sizes to perform symbolic burial, as well as to manage water flow.[59] Their migration transformed the landscape in the whole South America, even the old, occupied lowlands around central plateau caves, and civilizations overlap is likely to have happened.[31,32]

The Marajo Island was probably colonized once, around 3,100 y.a. At the Marajoara phase the island could have reached up to 1 million people,[60] with the summing up of villages and cities at the denser regions capable of accommodating up to 100,000 people.[57] The Marajoara phase was a complex society of mound constructors and pottery makers, with intense landscape management and urban structures that lasted from 1625 to 668 y.a. (400 to 1350 AD).[59] These human-modified, and organic matter-rich habitats could have become a substitute for the human-foxes cave habitats for *Lu. longipalpis*. Therefore, a recent adaptation of *Lu. longipalpis* to peri-urban and suburb habitats may never have happened from populations coming from the wild, but from insects ecologically adjusted to primitive human-modified habitats, previously found in the same regions where the species has been spreading now-a-days.

The system visceral leishmaniasis-canines-humans may have evolved separately and in the north Africa and Mediterranean Europe,[24] but a convergent phenomenon is possible to have happened in the Neotropical region.[61] In any case, for this host-parasite system to thrive, habi-

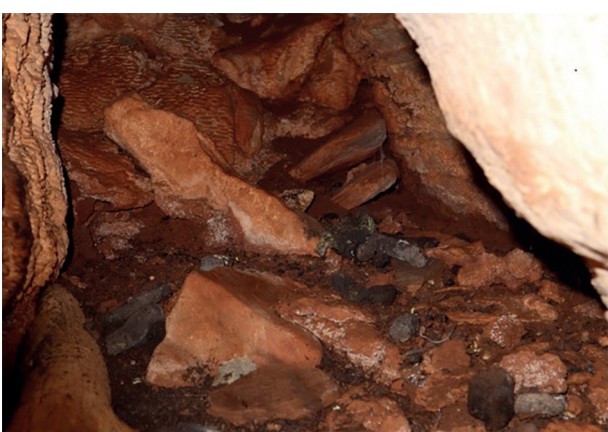

Fig. 4: surrounds of Lapinha Limestone cave, showing several fox dens or latrines, which are open to outside.

tat-specific insect vector species also should have to adapt to the habitats created by humans, as well as by canine packs. In both cases, heavily eutrophic dens, caves, and primitive human-made litter and ceremonial mounds, rich in feces and carcasses, may have shaped the environments around these vertebrate species (and also, may have been a main driver of canid domestication),[16] imposing specific selection on insects associated to such extreme conditions. There are reports on different Nations of Guarani ethnic groups who used to bury domesticated foxes of different species, highlighting the human-canid domestication relationship, which might happen to suffer hybridizations with European dogs later on.[15,34]

Ongoing research of our group has shown that *Lu. longipalpis* immatures fed on dog feces exhibit high ferritin (a decontaminating protein) transcription levels, similar to immatures exposed to iron-contaminated substrates. Hence, iron and other heavy metals found in organic matter such as carnivore feces and carcasses, act as a strong ecological filter, limiting sandfly diversity associated with canids. This is an avenue for future research.

In 1620, the Marajo Island was still heavily populated by native communities, but also threatened by a combination of Portuguese, Dutch and English invaders, based on both official forts and settlements, as well as pirates.[62] Mostly important, the Marajoara society (Nheengaíba nation) survived and resisted for 20 years a war promoted by the Portuguese.[62,63] Such war finished only when Jesuit missionaries were allowed into the island, promoting a partially pacific annexation, in 1659.[63]

The prolonged conflict likely created opportunities for VL infected domestic dogs to reach the island and get in contact with native foxes, in case introduction was indeed the pathway for the disease to spillover to native hosts. On the other hand, in the possibility of a native origin of *L. infantum*,[61] it is still possible to consider an evolutionary scenario of a four-species interaction, based on expected *L. infantum*'s niche demand. The conditions into which *L. infantum* spread and keep viable populations seemed to depend on: 1) highly contaminated and humid microhabitats prone to the insect vector domestication (likely to have been colonized by *Lu. longipalpis*, after adaptation to heavy metal contaminated organic matter); 2) densely human occupied and transformed landscapes.

The human-made habitats in South America during the late Holocene likely are convergent and similar to those in the Old World's hotspots for visceral leishmaniasis, namely, North Africa-South Europe, Mid-East and Southeast Asia.[17] All these regions are expected to have been densely populated and transformed in the late Holocene.[28,32] Such worldwide similarities in that stage of humanity evolution are also described as a baseline for diversification and dispersion of parasitic diseases, as well as canid domestication.[18,19]

In conclusion, our findings indicate that *Lu. longipalpis* may have adapted to living near humans long before European colonization. Recent archeological evidence suggests that original South American societies transformed dramatically the landscape, creating vast anthropomorphic environments,[64] likely to favor insect hematophagous as well as to create a landscape prone-to-transmission of diseases. The proximity of foxes to indigenous settlements may have further enhanced the creation of hyper-wet and nutrient-rich microhabitats, which are suitable to *Lu. longipalpis* immature development, around human occupied caves, as well as in primitive constructions in wetlands. Our species distribution data showed a very clear habitat-related co-occurrence of human-foxes-*Lu. Longipalpis*, providing solid, statistically supported, evidence for our conclusion.

The most relevant implication of our findings is that AVL, and also Old World VL, should be considered a long-term human related disease, associated with few sandfly species well adapted to our modified, and heavily contaminated, environments. Hence, and contrary to cutaneous leishmaniasis, this is not a sylvatic disease spilling-over to human societies. The narrative that the AVL invaded cities from wild ecosystems, under this perspective, should be revisited. *Lu. longipalpis* (as well as the *C. thous* fox), may have kept associated from old to new human settlements, especially in cave-rich biogeographic regions. Further studies on *Lu. longipalpis* immature development on habitats related to human-canid contaminated organic matter, as well as the identification of neglected urban habitats capable of retaining similar humidity and dirtiness like those in primitive caves, should be put in priority for disease control.

## ACKNOWLEDGEMENTS

To Instituto Estadual de Florestas do Estado de Minas Gerais for vital information and support in reaching the Minas Gerais preserved cave system; DC Jansen for all support in reaching the public data on Brazilian cave distribution; SIRGAS 2000; ICMBio/CECAV; ICMBio/Sistema de Avaliação do Risco de Extinção da Biodiversidade - SALVE for the availability of data.

## AUTHORS' CONTRIBUTION

SPR, CPD and NFG - conception; SPR, NFG, MHP and ABR - design of the work; SPR, CPD, RVD and MFBA - data acquisition and analysis; SPR, LGFM, NFG, CPD, JDAF, TDS and JGV - data interpretation; SPR - drafting the work; RVD,

MFBA, LGFM and TDS – figures; SPR, CEVG, NFG, MHP, ABR, JDAF, TDS and JGV - writing and theoretical review of the text. The authors declare no conflicts of interest. The contributions of the NIH authors were made as part of their official duties as NIH federal employees, and are in compliance with agency policy requirements, and are considered Works of the United States Government. However, the findings and conclusions presented in this paper are those of the authors and do not necessarily reflect the views of the NIH or the US Department of Health and Human Services.

## DATA AVAILABILITY

The data used in the analyses are available at https://drive.google.com/drive/folders/1VYyrtJr6dJXDGqhxdUSXRVfX5LZS1saG?.

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

# OPEN PEER REVIEW

Memórias do IOC thanks the anonymous reviewers for their contribution to the peer review of this work.

## FIRST REVIEW ROUND

REVIEWERS' COMMENTS

**REVIEWER #1**

In the study "Holocene man-occupied caves and transformed wetlands as a facilitating factor for Leishmania infantum in South America," the authors assume that during the Holocene (9000-2000 y.a.), human occupation of caves, as well as the continent's coastal and wetlands, combined with sedentism across a large area of South America, parallel to the domestication of foxes, led to the accumulation of organic matter that favored the development of immature L. longipalpis, the main vector of Leishmania infantum, the agent of visceral leishmaniasis, and favored the invasion of this agent introduced by Portuguese dogs into this continent. So, they argue that understanding the invasion process of L. infantum in the Americas can help understand the urbanization of the disease.

Therefore, a spatial study of the distribution of caves, the human population in the Holocene, and the current distribution of wild canids (foxes) and L. longipalpis was developed. They also investigated whether the presence of L. longipalpis was conditioned by biogeographical regions with abundant caves.

In background, there is sufficient and bibliographical information to inform the reader about the context in which the assumptions for development were based

Methods. The databases used for the analysis and the care taken to obtain information on L. longipalpis were clearly explained, as well as the statistical analyses used, which I consider appropriate for the approach.

The results were presented in descriptive and graphical form, showing the distribution of L. longipalpis with significant overlap of that humans in the Holocene and of the two fox species.

The distribution of the four species strongly suggests that coexistence occurred for a long period in many locations in central Brazil, with two hotspots in the Northeast region and the state of Minas Gerais.

They conclude that L. longipalpis underwent a process of adaptation to living near humans well before European colonization. The proximity of foxes to indigenous settlements may have fostered the formation of ultra-humid habitats rich in organic matter. The transformation of the environment by these primitive societies led to the creation of environments capable of sustaining hematophagous insects and their associated diseases.

I, therefore, consider that this novel approach taken in the study is quite intriguing and sheds light on understanding some of the determinants that led to the urbanization of visceral leishmaniasis in the Americas.

The research was conducted using appropriate methods for testing hypotheses and sufficiently referenced.

The results were well described and illustrated with the necessary graphs and figures.

The discussion was long, always well-supported by pertinent references and a coherent line of reasoning. They add another explanation to the relatively well-accepted hypothesis that the population of L. longipalpis that now occupies urban areas originated from wild areas. They believe that this species did indeed originate from the wild, having undergone an adaptive process to the anthropogenic environment during the Marajoara period of the island.

The conclusions were based on the results.

Regarding the abstract and discussion, I would like to suggest some modifications or complementation:

Abstract:

Please, revise the paragraph on methods, because is few confuse and add information about the statistical tests used for the analysis.

In the last paragraph, the phrase "how native humans and foxes were added as a host of L. infantum in South America" although presented from an ecological and biogeographical perspective, it may lead the reader to believe that primitive humans and foxes were infected by L. infantum.and its not possible, since the authors assumed that L. infantum invaded the Americas in colonial period.

Dscussion, the authors only considered the hypothesis that L. infantum invaded the American continent through infected Portuguese dogs during the colonial period. However, another hypothesis is raised in the study carried out by Silveira et al. (2023) regarding the origin of the leishmaniasis agent in the Americas. It would be interesting if the authors also considered this aspect, especially in the Marajoara period.

The reference to No. 27 is out of order.

Silveira et al. 2023 Comparative Genomic Analyses of New and Old World Viscerotropic Leishmanine Parasites: Further Insights into the Origins of Visceral Leishmaniasis Agents. Microorganisms 2023, 11, 25. https://doi.org/10.3390/ microorganisms11010025

**REVIEWER #2**

The manuscript is clearly written, presenting hypothesis, methods, results, and discussion in a coherent way. The interdisciplinary approach combining archaeology, human history, ecology, epidemiology, and biogeography is a strength of the work.

Major Comments

Terminology and Formatting: Standardize temporal references (e.g., years ago – y.a. or Before Present – BP), ensuring that the first mention appears in full with abbreviation in parentheses. Species names should follow taxonomic conventions: the first citation in full, thereafter abbreviated consistently (Lu. longipalpis and L. infantum).

Introduction: It is well-structured, with an interdisciplinary approach, covering archaeology, human history, ecology, evolution, epidemiology, and biogeography. This helps readers understand the general concepts of these fields, supported by good citations. Although extensive, I believe it is important to keep it due to the complexity of the subject.

Methodology: Clarify that municipalities with "zero records" may reflect lack of research rather than true species absence. To correlate the occurrence of Lu. longipalpis with the number of caves in the municipality, the sampling site of the sandfly was cross-checked with any actual cave occurrence within a radius of 100 km from the central coordinates of the sampling point. I have doubts whether 100 km might be too large a distance for correlating with the vector. Perhaps 1 km would be more appropriate, as with 100 km the probability of correlation increases significantly. I believe it is important to review this methodology in the spatial distribution analyses. Correct typographical errors (e.g., "5.5000" instead of "5,500").

Results: They are presented clearly and objectively, including validation of the main hypothesis, such as the association between the presence of Lu. longipalpis and municipalities with or near caves. Figures and labels should be standardized (e.g., avoid abbreviating Figure).

Discussion: Consider adding the hypothesis of iron acting as a toxic environmental filter, limiting sandfly diversity, as a potential avenue for future research. Also, restructure the discussion into continuous paragraphs, incorporating case studies such as human-modified late Holocene wetlands (e.g., Marajó Island). Include Migonemyia migonei as an additional vector species in the New World.

Conclusion: Expand the conclusion to highlight broader implications, potential research lines, and reinforce the statistical evidence supporting the central hypothesis.

Minor Issues

Ensure consistency in species abbreviations across the manuscript.

Recommendation

A valuable and well-prepared manuscript; it should be published. With the above revisions, it will be considerably strengthened and more suitable for publication.

<div align="right">AUTHORS' RESPONSE TO THE REVIEWERS</div>

Regarding the abstract and discussion, I would like to suggest some modifications or complementation:

Abstract:

Please, revise the paragraph on methods, because is few confuse and add information about the statistical tests used for the analysis.

Authors – First of all, thank you very much for your so positive and constructive criticism towards this paper. The description of methods in the Abstract was changed and expanded, within the limits imposed of 200 words for the whole Abstract. Hence, we also rearranged the remaining text for sake of clarity. Any new text is in red:

"BACKGROUND

South American humans transformed large extensions of the continent around cave ecosystems in the Holocene. Such transformations produced predictable eutrophic habitats that could have attracted foxes and further favored the insect vector of American Visceral Leishmaniosis (AVL), Lutzomyia longipalpis, to adapt to human-contaminated habitats.

OBJECTIVES

Here we present spatial analyses on the distribution of caves, Holocene human populations, the present-days main wild reservoirs of Leishmania infantum, Cerdocyon thous and Lycalopex vetulus, and the vector Lu. longipalpis in Brazil.

METHODS

The presence or absence of Lu. longipalpis in function of cave abundance, based on coordinates of all recorded samples (Fiocruz and GBIF database and literature), and cave locations taken from ICMBio/CECAV database, were tested by Contingency Table. The overlap in the distribution of Lu. longipalpis, C. thous and L. vetulus with humans from Holocene was tested by PERMANOVA from a NMDS, using published archaeological data on human demography and ICMBio´s data on caves and foxes.

FINDINGS

Caves are present in 18% of Brazilian municipalities, and Lu. longipalpis were significantly more frequent in these places than in municipalities without caves. Native humans and foxes have broader distributions than caves but co-occurred with Lu. longipalpis in cave-ecosystems.

MAIN CONCLUSION

The most relevant implication of our findings is that AVL should be considered a long-term human related disease, associated with few sandfly species well adapted to our modified, and heavily contaminated, environments."

In the last paragraph, the phrase "how native humans and foxes were added as a host of L. infantum in South America" although presented from an ecological and biogeographical perspective, it may lead the reader to believe that primitive humans and foxes were infected by L. infantum and its not possible, since the authors assumed that L. infantum invaded the Americas in colonial period.

Authors – the sentence was changed, thank you.

Discussion, the authors only considered the hypothesis that L. infantum invaded the American continent through infected Portuguese dogs during the colonial period. However, another hypothesis is raised in the study carried out by Silveira et al. (2023) regarding the origin of the leishmaniasis agent in the Americas. It would be interesting if the authors also considered this aspect, especially in the Marajoara period.

Authors – Than you for this suggestion and to point out such amazing article. We added a paragraph considering the hypothesis of a native L. infantum scenario, and how that could be considered into our ecological niche approach for a pre-colonial parasitic environment that, due to humans, should have been favourable to AVL from the start. The paragraph is in red, from line 403 to 417 by the end of the discussion:

"The prolonged conflict likely created opportunities for VL infected domestic dogs to reach the island and get in contact with native foxes, in case introduction was indeed the pathway for the disease to spillover to native hosts. On the other hand, in the possibility of a native origin of L. infantum,(61) it is still possible to consider an evolutionary scenario of a four-species interaction, based on expected L. infantum´s niche demand. The conditions into which L. infantum spread and keep viable populations seemed to depend on: 1) highly contaminated and humid microhabitats prone to the insect vector domestication (likely to have been colonized by Lu. longipalpis, after adaptation to a heavy metal contaminated organic matter); 2) densely human occupied and transformed landscapes.

The human-made habitats South America during the late Holocene likely are convergent and similar to those in the Old World´s hotspots for visceral leishmaniasis, namely, North Africa-South Europe, Mid-East and Southeast Asia.(17) All these regions are expected to have been densely populated and transformed in the late Holocene.(28,32) Such worldwide similarities in that stage of humanity evolution are also described as a baseline for diversification and dispersion of parasitic diseases, as well as canid domestication.(18,19)"

The reference to No. 27 is out of order.

Authors – Corrected, thank you.

Silveira et al. 2023 Comparative Genomic Analyses of New and Old World Viscerotropic Leishmanine Parasites: Further Insights into the Origins of Visceral Leishmaniasis Agents. Microorganisms 2023, 11, 25. https://doi.org/10.3390/ microorganisms11010025

Reviewer: 2

The manuscript is clearly written, presenting hypothesis, methods, results, and discussion in a coherent way. The interdisciplinary approach combining archaeology, human history, ecology, epidemiology, and biogeography is a strength of the work.

Major Comments

Terminology and Formatting: Standardize temporal references (e.g., years ago – y.a. or Before Present – BP), ensuring that the first mention appears in full with abbreviation in parentheses. Species names should follow taxonomic conventions: the first citation in full, thereafter abbreviated consistently (Lu. longipalpis and L. infantum).

Authors – all checked and corrected, thank you.

Introduction: It is well-structured, with an interdisciplinary approach, covering archaeology, human history, ecology, evolution, epidemiology, and biogeography. This helps readers understand the general concepts of these fields, supported by good citations. Although extensive, I believe it is important to keep it due to the complexity of the subject.

Methodology: Clarify that municipalities with "zero records" may reflect lack of research rather than true species absence.

Authors – Indeed, and thank you. We had mentioned this, and actually we validated the insect distribution by the SINAN´s notifications on AVL. A more explicit sentence on this was added in lines 232-234:

"Because a reasonable proportion of zeros might be due to lack of research rather than actual absence of the species, we validated the distribution of Lu. longipalpis, and verify the effect of possible under sampling, we overlap the distribution of the insect with the distribution of AVL incidence of cases in the Country, taken from the Information System of Injuries Notification (SINAN), Ministry of Health, from 2001 to 2021."

To correlate the occurrence of Lu. longipalpis with the number of caves in the municipality, the sampling site of the sandfly was cross-checked with any actual cave occurrence within a radius of 100 km from the central coor-

dinates of the sampling point. I have doubts whether 100 km might be too large a distance for correlating with the vector. Perhaps 1 km would be more appropriate, as with 100 km the probability of correlation increases significantly. I believe it is important to review this methodology in the spatial distribution analyses.

Authors – Concerning this enquiring, we would agree if the analysis used aimed to test a direct association of L. longipalpis with the caves. Nevertheless, we tested whether specimens belonged to a cave ecosystem, thus exploring the region scale and not the sample location. One basic reason for that is the urbanization of this species at the present. As most samples were taken within or too close to cities, the proximity of an actual cave would be unlikely to be detected or to be relevant at the present ecological scenario. However, we expect that cities within a cave ecosystem would be more vulnerable than others, and this correction was applied only to prevent bias caused by political limits of the municipality. We better explained this in lines 220-226, In the methods:

"This area size was defined arbitrarily, but in order to deal with the wide range of municipality sizes in the Country (from 3.5 to 60,431 Km2 in Southeast and Northeast regions - https://www.ibge.gov.br/geociencias/). However, this correction was made only to prevent any underestimation of cave density in situations in which a Lu. longipalpis sample could have been taken in a municipality out of core distribution of caves, but still close enough to cavities."

Correct typographical errors (e.g., "5.5000" instead of "5,500").

Authors – all checked and corrected, thank you.

Results: They are presented clearly and objectively, including validation of the main hypothesis, such as the association between the presence of Lu. longipalpis and municipalities with or near caves. Figures and labels should be standardized (e.g., avoid abbreviating Figure).

Authors – all checked and corrected, thank you.

Discussion: Consider adding the hypothesis of iron acting as a toxic environmental filter, limiting sandfly diversity, as a potential avenue for future research.

Authors – in lines 388-393 we added the following sentence, explaining our ongoing findings about iron as an ecological filter, although we also mentioned it in the first version, in the second paragraph of the discussion:

"Ongoing research of our group has shown that Lu. longipalpis immatures fed on dog feces exhibit high ferritin (a decontaminating protein) transcription levels, similar to immatures exposed to iron-contaminated substrates. Hence, iron and other heavy metals found in organic matter such as carnivore feces and carcasses, act as a strong ecological filter, limiting sandfly diversity associated to canids. This is an avenue for future research."

Also, restructure the discussion into continuous paragraphs, incorporating case studies such as human-modified late Holocene wetlands (e.g., Marajó Island).

Authors – Did, thank you.

Include Migonemyia migonei as an additional vector species in the New World.

Authors – Included in line 296, where we mentioned other putative vectors, thank you for the reminding about this species.

Conclusion: Expand the conclusion to highlight broader implications, potential research lines, and reinforce the statistical evidence supporting the central hypothesis.

Authors –Conclusion was fully modified to attend the reviewer requests. Thank you:

"Our findings indicate that Lu. longipalpis may have adapted to living near humans long before European colonization. Recent archeological evidence suggests that original South American societies transformed dramatically the landscape, creating vast anthropomorphic environments,(64) likely to favor insect hematophagous as well as to create a landscape prone-to-transmission of diseases. The proximity of foxes to indigenous settlements may have further enhanced the creation of hyper-wet and nutrient-rich microhabitats, which are suitable to Lu. longipalpis immature development, around human occupied caves, as well as in primitive constructions in wetlands. Our species distribution data showed a very clear habitat-related co-occurrence of human-foxes-Lu. Longipalpis, providing solid, statistically supported, evidence for our conclusion.

The most relevant implication of our findings is that AVL, and also Old World VL, should be considered a long-term human related disease, associated with few sandfly species well adapted to our modified, and heavily contaminated, environments. Hence, and contrary to cutaneous leishmaniasis, this is not a sylvatic disease spilling-over to human societies. The narrative that the AVL invaded cities from wild ecosystems, under this perspective, should be revisited. Lutzomyia longipalpis (as well as the C. thous fox), may have kept associated from old to new human settlements, especially in cave-rich biogeographic regions. Further studies on Lu. longipalpis immature development on habitats related to human-canid contaminated organic matter, as well as the identification of neglected urban habitats capable of retaining similar humidity and dirtiness like those in primitive caves, should be put in priority for the disease control."

Minor Issues

Ensure consistency in species abbreviations across the manuscript.

Authors – Did, thank you.

Recommendation

A valuable and well-prepared manuscript; it should be published. With the above revisions, it will be considerably strengthened and more suitable for publication.

## SECOND REVIEW ROUND

REVIEWERS' COMMENTS

**REVIEWER #1**

All aspects related to this item are adequately covered.

I only ask that the authors consider replacing the word domestication with domiciliation in line 407, and changing Longipalpis to longipalpis in line 428.

**REVIEWER #2**

No comments.

