## [Reviewer Report · FIRST REVIEW ROUND - REVIEWERS COMMENTS]

## REVIEWER #1

In the study “Holocene man-occupied caves and transformed wetlands as a facilitating factor for Leishmania infantum in South America,” the authors assume that during the Holocene (9000-2000 y.a.), human occupation of caves, as well as the continent’s coastal and wetlands, combined with sedentism across a large area of South America, parallel to the domestication of foxes, led to the accumulation of organic matter that favored the development of immature L. longipalpis, the main vector of Leishmania infantum, the agent of visceral leishmaniasis, and favored the invasion of this agent introduced by Portuguese dogs into this continent. So, they argue that understanding the invasion process of L. infantum in the Americas can help understand the urbanization of the disease. 

Therefore, a spatial study of the distribution of caves, the human population in the Holocene, and the current distribution of wild canids (foxes) and L. longipalpis was developed. They also investigated whether the presence of L. longipalpis was conditioned by biogeographical regions with abundant caves. 

In background, there is sufficient and bibliographical information to inform the reader about the context in which the assumptions for development were based

Methods. The databases used for the analysis and the care taken to obtain information on L. longipalpis were clearly explained, as well as the statistical analyses used, which I consider appropriate for the approach.

The results were presented in descriptive and graphical form, showing the distribution of L. longipalpis with significant overlap of that humans in the Holocene and of the two fox species.

The distribution of the four species strongly suggests that coexistence occurred for a long period in many locations in central Brazil, with two hotspots in the Northeast region and the state of Minas Gerais.

They conclude that L. longipalpis underwent a process of adaptation to living near humans well before European colonization. The proximity of foxes to indigenous settlements may have fostered the formation of ultra-humid habitats rich in organic matter. The transformation of the environment by these primitive societies led to the creation of environments capable of sustaining hematophagous insects and their associated diseases.

I, therefore, consider that this novel approach taken in the study is quite intriguing and sheds light on understanding some of the determinants that led to the urbanization of visceral leishmaniasis in the Americas.

The research was conducted using appropriate methods for testing hypotheses and sufficiently referenced.

The results were well described and illustrated with the necessary graphs and figures.

The discussion was long, always well-supported by pertinent references and a coherent line of reasoning. They add another explanation to the relatively well-accepted hypothesis that the population of L. longipalpis that now occupies urban areas originated from wild areas. They believe that this species did indeed originate from the wild, having undergone an adaptive process to the anthropogenic environment during the Marajoara period of the island.

The conclusions were based on the results.

Regarding the abstract and discussion, I would like to suggest some modifications or complementation:

Abstract:

Please, revise the paragraph on methods, because is few confuse and add information about the statistical tests used for the analysis.

In the last paragraph, the phrase “how native humans and foxes were added as a host of L. infantum in South America” although presented from an ecological and biogeographical perspective, it may lead the reader to believe that primitive humans and foxes were infected by L. infantum.and its not possible, since the authors assumed that L. infantum invaded the Americas in colonial period.

Dscussion, the authors only considered the hypothesis that L. infantum invaded the American continent through infected Portuguese dogs during the colonial period. However, another hypothesis is raised in the study carried out by Silveira et al. (2023) regarding the origin of the leishmaniasis agent in the Americas. It would be interesting if the authors also considered this aspect, especially in the Marajoara period.

The reference to No. 27 is out of order.

Silveira et al. 2023 Comparative Genomic Analyses of New and Old World Viscerotropic Leishmanine Parasites: Further Insights into the Origins of Visceral Leishmaniasis Agents. Microorganisms 2023, 11, 25. https://doi.org/10.3390/ microorganisms11010025

## REVIEWER #2

The manuscript is clearly written, presenting hypothesis, methods, results, and discussion in a coherent way. The interdisciplinary approach combining archaeology, human history, ecology, epidemiology, and biogeography is a strength of the work.

Major Comments

Terminology and Formatting: Standardize temporal references (e.g., years ago – y.a. or Before Present – BP), ensuring that the first mention appears in full with abbreviation in parentheses. Species names should follow taxonomic conventions: the first citation in full, thereafter abbreviated consistently (Lu. longipalpis and L. infantum).

Introduction: It is well-structured, with an interdisciplinary approach, covering archaeology, human history, ecology, evolution, epidemiology, and biogeography. This helps readers understand the general concepts of these fields, supported by good citations. Although extensive, I believe it is important to keep it due to the complexity of the subject.

Methodology: Clarify that municipalities with “zero records” may reflect lack of research rather than true species absence. To correlate the occurrence of Lu. longipalpis with the number of caves in the municipality, the sampling site of the sandfly was cross-checked with any actual cave occurrence within a radius of 100 km from the central coordinates of the sampling point. I have doubts whether 100 km might be too large a distance for correlating with the vector. Perhaps 1 km would be more appropriate, as with 100 km the probability of correlation increases significantly. I believe it is important to review this methodology in the spatial distribution analyses. Correct typographical errors (e.g., “5.5000” instead of “5,500”). 

Results: They are presented clearly and objectively, including validation of the main hypothesis, such as the association between the presence of Lu. longipalpis and municipalities with or near caves. Figures and labels should be standardized (e.g., avoid abbreviating Figure).

Discussion: Consider adding the hypothesis of iron acting as a toxic environmental filter, limiting sandfly diversity, as a potential avenue for future research. Also, restructure the discussion into continuous paragraphs, incorporating case studies such as human-modified late Holocene wetlands (e.g., Marajó Island). Include Migonemyia migonei as an additional vector species in the New World. 

Conclusion: Expand the conclusion to highlight broader implications, potential research lines, and reinforce the statistical evidence supporting the central hypothesis.

Minor Issues

Ensure consistency in species abbreviations across the manuscript.

Recommendation

A valuable and well-prepared manuscript; it should be published. With the above revisions, it will be considerably strengthened and more suitable for publication.

---

## [Author Response · AUTHORS RESPONSE TO REVIEWERS]

## Regarding the abstract and discussion, I would like to suggest some modifications or complementation:

Abstract:

Please, revise the paragraph on methods, because is few confuse and add information about the statistical tests used for the analysis.

Authors – First of all, thank you very much for your so positive and constructive criticism towards this paper. The description of methods in the Abstract was changed and expanded, within the limits imposed of 200 words for the whole Abstract. Hence, we also rearranged the remaining text for sake of clarity. Any new text is in red:

“BACKGROUND South American humans transformed large extensions of the continent around cave ecosystems in the Holocene. Such transformations produced predictable eutrophic habitats that could have attracted foxes and further favored the insect vector of American Visceral Leishmaniosis (AVL), Lutzomyia longipalpis, to adapt to human-contaminated habitats.

OBJECTIVES Here we present spatial analyses on the distribution of caves, Holocene human populations, the present-days main wild reservoirs of Leishmania infantum, Cerdocyon thous and Lycalopex vetulus, and the vector Lu. longipalpis in Brazil.

METHODS The presence or absence of Lu. longipalpis in function of cave abundance, based on coordinates of all recorded samples (Fiocruz and GBIF database and literature), and cave locations taken from ICMBio/CECAV database, were tested by Contingency Table. The overlap in the distribution of Lu. longipalpis, C. thous and L. vetulus with humans from Holocene was tested by PERMANOVA from a NMDS, using published archaeological data on human demography and ICMBio´s data on caves and foxes.

FINDINGS Caves are present in 18% of Brazilian municipalities, and Lu. longipalpis were significantly more frequent in these places than in municipalities without caves. Native humans and foxes have broader distributions than caves but co-occurred with Lu. longipalpis in cave-ecosystems.

MAIN CONCLUSION The most relevant implication of our findings is that AVL should be considered a long-term human related disease, associated with few sandfly species well adapted to our modified, and heavily contaminated, environments.” 

In the last paragraph, the phrase “how native humans and foxes were added as a host of L. infantum in South America” although presented from an ecological and biogeographical perspective, it may lead the reader to believe that primitive humans and foxes were infected by L. infantum and its not possible, since the authors assumed that L. infantum invaded the Americas in colonial period.

Authors – the sentence was changed, thank you.

Discussion, the authors only considered the hypothesis that L. infantum invaded the American continent through infected Portuguese dogs during the colonial period. However, another hypothesis is raised in the study carried out by Silveira et al. (2023) regarding the origin of the leishmaniasis agent in the Americas. It would be interesting if the authors also considered this aspect, especially in the Marajoara period.

Authors – Than you for this suggestion and to point out such amazing article. We added a paragraph considering the hypothesis of a native L. infantum scenario, and how that could be considered into our ecological niche approach for a pre-colonial parasitic environment that, due to humans, should have been favourable to AVL from the start. The paragraph is in red, from line 403 to 417 by the end of the discussion:

“The prolonged conflict likely created opportunities for VL infected domestic dogs to reach the island and get in contact with native foxes, in case introduction was indeed the pathway for the disease to spillover to native hosts. On the other hand, in the possibility of a native origin of L. infantum,(61) it is still possible to consider an evolutionary scenario of a four-species interaction, based on expected L. infantum´s niche demand. The conditions into which L. infantum spread and keep viable populations seemed to depend on: 1) highly contaminated and humid microhabitats prone to the insect vector domestication (likely to have been colonized by Lu. longipalpis, after adaptation to a heavy metal contaminated organic matter); 2) densely human occupied and transformed landscapes.

The human-made habitats South America during the late Holocene likely are convergent and similar to those in the Old World´s hotspots for visceral leishmaniasis, namely, North Africa-South Europe, Mid-East and Southeast Asia.(17) All these regions are expected to have been densely populated and transformed in the late Holocene.(28,32) Such worldwide similarities in that stage of humanity evolution are also described as a baseline for diversification and dispersion of parasitic diseases, as well as canid domestication.(18,19)” 

The reference to No. 27 is out of order.

Authors – Corrected, thank you.

Silveira et al. 2023 Comparative Genomic Analyses of New and Old World Viscerotropic Leishmanine Parasites: Further Insights into the Origins of Visceral Leishmaniasis Agents. Microorganisms 2023, 11, 25. https://doi.org/10.3390/ microorganisms11010025

## Reviewer #2

The manuscript is clearly written, presenting hypothesis, methods, results, and discussion in a coherent way. The interdisciplinary approach combining archaeology, human history, ecology, epidemiology, and biogeography is a strength of the work.

Major Comments

Terminology and Formatting: Standardize temporal references (e.g., years ago – y.a. or Before Present – BP), ensuring that the first mention appears in full with abbreviation in parentheses. Species names should follow taxonomic conventions: the first citation in full, thereafter abbreviated consistently (Lu. longipalpis and L. infantum).

Authors – all checked and corrected, thank you.

Introduction: It is well-structured, with an interdisciplinary approach, covering archaeology, human history, ecology, evolution, epidemiology, and biogeography. This helps readers understand the general concepts of these fields, supported by good citations. Although extensive, I believe it is important to keep it due to the complexity of the subject.

Methodology: Clarify that municipalities with “zero records” may reflect lack of research rather than true species absence.

Authors – Indeed, and thank you. We had mentioned this, and actually we validated the insect distribution by the SINAN´s notifications on AVL. A more explicit sentence on this was added in lines 232-234:

“Because a reasonable proportion of zeros might be due to lack of research rather than actual absence of the species, we validated the distribution of Lu. longipalpis, and verify the effect of possible under sampling, we overlap the distribution of the insect with the distribution of AVL incidence of cases in the Country, taken from the Information System of Injuries Notification (SINAN), Ministry of Health, from 2001 to 2021.”

To correlate the occurrence of Lu. longipalpis with the number of caves in the municipality, the sampling site of the sandfly was cross-checked with any actual cave occurrence within a radius of 100 km from the central coordinates of the sampling point. I have doubts whether 100 km might be too large a distance for correlating with the vector. Perhaps 1 km would be more appropriate, as with 100 km the probability of correlation increases significantly. I believe it is important to review this methodology in the spatial distribution analyses.

Authors – Concerning this enquiring, we would agree if the analysis used aimed to test a direct association of L. longipalpis with the caves. Nevertheless, we tested whether specimens belonged to a cave ecosystem, thus exploring the region scale and not the sample location. One basic reason for that is the urbanization of this species at the present. As most samples were taken within or too close to cities, the proximity of an actual cave would be unlikely to be detected or to be relevant at the present ecological scenario. However, we expect that cities within a cave ecosystem would be more vulnerable than others, and this correction was applied only to prevent bias caused by political limits of the municipality. We better explained this in lines 220-226, In the methods: “This area size was defined arbitrarily, but in order to deal with the wide range of municipality sizes in the Country (from 3.5 to 60,431 Km2 in Southeast and Northeast regions - https://www.ibge.gov.br/geociencias/). However, this correction was made only to prevent any underestimation of cave density in situations in which a Lu. longipalpis sample could have been taken in a municipality out of core distribution of caves, but still close enough to cavities.” 

Correct typographical errors (e.g., “5.5000” instead of “5,500”).

Authors – all checked and corrected, thank you.

Results: They are presented clearly and objectively, including validation of the main hypothesis, such as the association between the presence of Lu. longipalpis and municipalities with or near caves. Figures and labels should be standardized (e.g., avoid abbreviating Figure).

Authors – all checked and corrected, thank you.

Discussion: Consider adding the hypothesis of iron acting as a toxic environmental filter, limiting sandfly diversity, as a potential avenue for future research.

Authors – in lines 388-393 we added the following sentence, explaining our ongoing findings about iron as an ecological filter, although we also mentioned it in the first version, in the second paragraph of the discussion: “Ongoing research of our group has shown that Lu. longipalpis immatures fed on dog feces exhibit high ferritin (a decontaminating protein) transcription levels, similar to immatures exposed to iron-contaminated substrates. Hence, iron and other heavy metals found in organic matter such as carnivore feces and carcasses, act as a strong ecological filter, limiting sandfly diversity associated to canids. This is an avenue for future research.” 

Also, restructure the discussion into continuous paragraphs, incorporating case studies such as human-modified late Holocene wetlands (e.g., Marajó Island).

Authors – Did, thank you.

Include Migonemyia migonei as an additional vector species in the New World.

Authors – Included in line 296, where we mentioned other putative vectors, thank you for the reminding about this species.

Conclusion: Expand the conclusion to highlight broader implications, potential research lines, and reinforce the statistical evidence supporting the central hypothesis.

Authors –Conclusion was fully modified to attend the reviewer requests. Thank you: “Our findings indicate that Lu. longipalpis may have adapted to living near humans long before European colonization. Recent archeological evidence suggests that original South American societies transformed dramatically the landscape, creating vast anthropomorphic environments,(64) likely to favor insect hematophagous as well as to create a landscape prone-to-transmission of diseases. The proximity of foxes to indigenous settlements may have further enhanced the creation of hyper-wet and nutrient-rich microhabitats, which are suitable to Lu. longipalpis immature development, around human occupied caves, as well as in primitive constructions in wetlands. Our species distribution data showed a very clear habitat-related co-occurrence of human-foxes-Lu. Longipalpis, providing solid, statistically supported, evidence for our conclusion. The most relevant implication of our findings is that AVL, and also Old World VL, should be considered a long-term human related disease, associated with few sandfly species well adapted to our modified, and heavily contaminated, environments. Hence, and contrary to cutaneous leishmaniasis, this is not a sylvatic disease spilling-over to human societies. The narrative that the AVL invaded cities from wild ecosystems, under this perspective, should be revisited. Lutzomyia longipalpis (as well as the C. thous fox), may have kept associated from old to new human settlements, especially in cave-rich biogeographic regions. Further studies on Lu. longipalpis immature development on habitats related to human-canid contaminated organic matter, as well as the identification of neglected urban habitats capable of retaining similar humidity and dirtiness like those in primitive caves, should be put in priority for the disease control.”

Minor Issues

Ensure consistency in species abbreviations across the manuscript.

Authors – Did, thank you.

Recommendation

A valuable and well-prepared manuscript; it should be published. With the above revisions, it will be considerably strengthened and more suitable for publication.

---

## [Reviewer Report · REVIEWERS COMMENTS]

## REVIEWER #1

All aspects related to this item are adequately covered.

I only ask that the authors consider replacing the word domestication with domiciliation in line 407, and changing Longipalpis to longipalpis in line 428.

## REVIEWER #2

No comments.